# MASTER: Multi-task Pre-trained Bottlenecked Masked Autoencoders are Better Dense Retrievers

## Abstract

Dense retrieval aims to map queries and passages into low-dimensional vector space for efficient similarity measuring, showing promising effectiveness in various large-scale retrieval tasks. Since most existing methods commonly adopt pre-trained Transformers (*e.g.,* BERT) for parameter initialization, some work focuses on proposing new pre-training tasks for compressing the useful semantic information from passages into dense vectors, achieving remarkable performances. However, it is still challenging to effectively capture the rich semantic information and relations about passages into the dense vectors via one single particular pre-training task. In this work, we propose a multi-task pre-trained model, MASTER, that unifies and integrates multiple pre-training tasks with different learning objectives under the bottlenecked masked autoencoder architecture. Concretely, MASTER utilizes a multi-decoder architecture to integrate three types of pre-training tasks: corrupted passages recovering, related passage recovering and PLMs outputs recovering. By incorporating a shared deep encoder, we construct a representation bottleneck in our architecture, compressing the abundant semantic information across tasks into dense vectors. The first two types of tasks concentrate on capturing the semantic information of passages and relationships among them within the pre-training corpus. The third one can capture the knowledge beyond the corpus from external PLMs (*e.g.,* GPT-2). Extensive experiments on several large-scale passage retrieval datasets have shown that our approach outperforms the previous state-of-the-art dense retrieval methods.

## 1 Introduction

Recent years have witnessed the great success of dense retrieval methods (Karpukhin et al., 2020; Qu et al., 2021; Xiong et al., 2021) in industrial applications, *e.g.,* web search (Brickley et al., 2019; Qiu et al., 2022) and question answering (Karpukhin et al., 2020; Izacard & Grave, 2021). These methods typically encode queries and passages into low-dimensional dense vectors and utilize the vector similarity between them to measure semantic relevance. In real-world applications, the dense vectors of large amounts of passages will be pre-computed. Then the approximate nearest neighbor (ANN) search techniques (Johnson et al., 2021) can be incorporated for efficient retrieval.

To generate high-quality dense vectors, pre-trained language models (PLMs) (Devlin et al., 2019; Liu et al., 2019) have been widely adopted as the backbone of the query and passage encoders. However, general PLMs (*e.g.,* BERT (Devlin et al., 2019)) may not be the best for dense retrieval, as their produced native dense representations (usually the `[CLS]` embedding) are not designed on purpose to generalize the information from the input text. To solve it, recent studies (Gao & Callan, 2021a; Lu et al., 2021; Sachan et al., 2021) adopt pre-training techniques to endow the `[CLS]` embedding the capacity of compressing the semantic information of the input text. They either rely on the autoencoding task that utilizes the `[CLS]` embedding to recover the corrupted text (*e.g.,* masked or replaced tokens) (Liu & Shao, 2022; Wang et al., 2022; Wu et al., 2022), or leverage the contrastive learning objective to capture the relations among passages (*e.g.,* co-occurrence) (Ram et al., 2022; Sachan et al., 2021), outperforming general PLMs in this task.

Despite the success, it is obvious that neither the autoencoding nor contrastive learning pre-training task is optimal to fully exploit the useful characteristics into the dense embedding for the retrieval task, as either mainly relies on limited specific information or relation from the corpus. From this point of view, adopting the multi-task pre-training framework that jointly learns various supervised signals from different tasks is promising. However, due to the divergences of input formats and learning objectives among different tasks, an arbitrary integration of these tasks is inappropriate, which may even cause detrimental gradient interference, leading to performance degradation (Kendall et al., 2018; Yu et al., 2020).

To address this problem, we consider integrating multiple pre-training tasks in a unified format and reducing the divergence of different training objectives. Since most of the NLP tasks can be reformulated as the text-to-text format (Xie et al., 2022; Raffel et al., 2020), we can also reconstruct the available pre-training tasks into such a format. Recently, the idea of bottlenecked masked autoencoder (BMAE) (Liu & Shao, 2022; Wang et al., 2022; Wu et al., 2022) has been proposed to pre-train dense retrievers, which typically adopts an encoder-decoder architecture, consisting of a deep encoder to generate the dense vector of the input texts and a shallow decoder that relies on the dense vector to recover an aggressively-masked text. In this way, an information bottleneck is constructed. The deep encoder should force the dense vector to reserve as much useful information as possible that is beneficial for recovering the text in the shallow decoder. Inspired by it, we consider unifying multiple different pre-training tasks into the BMAE format, *i.e.,* taking texts as the input of the encoder and recovering itself or its related text in the decoder. For example, to capture the passage relations (*e.g.,* co-occurrence), we can utilize a passage as the encoder's input and leverage its dense vector to help recover an aggressive masked related passage in the decoder. Such a unified way is promising to solve the central issue of the multi-task pre-training derived from the divergences of input formats and learning objectives among different tasks, and capture a variety of semantics or relations in different tasks to pre-train effective dense vectors.

Based on the above motivation, we propose **MASTER**, a **m**ulti-t**as**k pre-**t**rained bottlenecked masked auto**e**ncode**r**, that adopts an multi-decoder architecture to integrate diverse pre-training tasks in the BMAE format. For each pre-training task, we devise a task-specific decoder to accomplish it, and all these decoders should rely on the output dense vector from the shared deep encoder to guide the decoding process. In this way, we construct multiple information bottlenecks to enforce the deep encoder to generate more informative dense vectors, leading to compressed high-quality representations. To learn sufficient useful semantics and relations, we devise three types of pre-training tasks: corrupted passages recovering, related passages recovering, and PLMs outputs recovering, respectively, a total of five tasks for pre-training. The first two types of tasks focus on compressing the semantic information of passages and modeling the relationships among passages within the corpus. The third type of tasks forces the dense vector of the input passage to recover the output text from other public generative PLMs like GPT-2 (Radford et al., 2019), which are capable of capturing the semantic information and relations beyond the corpus to further enhance the dense vectors.

To verify the effectiveness of our approach, we conduct extensive experiments on several text retrieval datasets, *e.g.,* MS-MARCO Passage Ranking (Nguyen et al., 2016), TREC Deep Learning Track (Craswell et al., 2020; 2021), Natural Questions (Kwiatkowski et al., 2019) and BEIR zero-shot retrieval benchmark Thakur et al. (2021). Experimental results show that our approach can achieve new state-of-the-art performances in dense retrieval. We will make the code and model checkpoints publicly available.

## 2 RELATED WORK

**Dense Retrieval.** Recent years have witnessed the remarkable progress of dense retrieval (Karpukhin et al., 2020; Zhan et al., 2020; Hong et al., 2022; Ram et al., 2022). Different from traditional sparse retrieval methods (*e.g.,* BM25 (Robertson et al., 2009)), dense retrieval approaches typically map queries and documents into low-dimensional dense vectors via a dual-encoder architecture and then utilize vector distance metrics (*e.g.,* dot product and cosine similarity) as the relevance scores. Such a way is supported by the efficient approximate nearest neighbor (ANN) search engines, *e.g.,* FAISS (Johnson et al., 2021). For effectively training dense retrieval models, existing work typically leverages pre-trained Transformers (Liu et al., 2019; Devlin et al., 2019) to initialize the dual encoders and then samples high-quality negatives when fine-tuning the encoders with con-

trastive learning objectives. Early work (Karpukhin et al., 2020; Min et al., 2020) mainly relies on in-batch random negatives and hard negatives mined by BM25. Afterward, a line of work (Qu et al., 2021; Xiong et al., 2021) finds that sampling top-$k$ ranked documents by a trained dense retriever as hard negatives is more beneficial for fine-tuning. However, a common problem for such top-$k$ negative sampling strategies is that they are easy to select false negatives, which impedes better performances. To alleviate the problem, current studies have explored several practical directions, including knowledge distillation with rerankers (Qu et al., 2021; Ren et al., 2021b; Lu et al., 2022), more effective pre-training methods (Zhou et al., 2022; Xu et al., 2022), denoising techniques (Mao et al., 2022; Hofstätter et al., 2021) and so on.

**Pre-training for Dense Retrieval.** PLMs (Devlin et al., 2019; Sanh et al., 2019; Sun et al., 2020) have become the de facto backbone of NLP models for their remarkable performances on various tasks. However, the fact is that these models are pre-trained with general tasks (*e.g.,* masked language model) without any prior task knowledge. Therefore, existing work finds that they are not ready to use for dense retrieval (Gao & Callan, 2021a; 2022), especially in low-data situations (Xu et al., 2022). To solve this issue, several studies (Gao & Callan, 2021a; Lu et al., 2021; Xu et al., 2022) are proposed to make the output sentence embedding more informative and discriminative. A type of work relies on the explicit relations between text pairs and designs the pre-training tasks based on the contrastive learning objective (Lee et al., 2019; Chang et al., 2020; Ma et al., 2022), *e.g.,* inverse cloze task and contrastive span prediction. Another line of work focuses on compressing the textual semantic information into the [CLS] embedding. These methods prefer to leverage the masked autoencoder architecture that incorporates a deep encoder and a shallow decoder, forcing the [CLS] embedding of the input text from the encoders to recover itself (Liu & Shao, 2022; Wu et al., 2022) or related texts (Wang et al., 2022).

## 3   PRELIMINARY

In this section, we introduce the task definition of this work and present the typical fine-tuning process of dense retrieval.

**Task Definition.** Given a query $q$, the dense passage retrieval task aims to retrieve the most relevant top-$k$ passages $\{p_i\}_{i=1}^k$ from a large candidate pool $\mathcal{P}$. To achieve this goal, the dual-encoder architecture is widely used. It consists of a query encoder $E_q$ and a passage encoder $E_p$, mapping the query $q$ and passage $p$ into $k$-dimensional dense vectors $\mathbf{h}_q$ and $\mathbf{h}_p$, respectively. Then, the semantic relevance score of $q$ and $p$ will be computed using dot product as

$$s(q, p) = \mathbf{h}_q \cdot \mathbf{h}_p. \tag{1}$$

Existing work mostly adopts pre-trained Transformers (*e.g.,* BERT (Devlin et al., 2019)) as the two encoders, using the representations of the [CLS] token as the dense vectors. In this work, we aim to propose a more effective multi-task pre-training framework specially for the dense retrieval task, which learns to compress more useful information into the [CLS] representations. Formally, given a pre-training corpus and a Transformer encoder, we focus on devising several tasks to pre-train the parameters of it. Then, the pre-trained Transformer will be used as the backbone of the query encoder $E_q$ and passage encoder $E_p$, and can be fine-tuned on downstream dense retrieval tasks.

**Fine-tuning Dense Retrievers.** In the fine-tuning stage, the learning objective is to pull the representations of a query $q$ and its relevant passages $\mathcal{P}^+$ together (as positives), while pushing apart irrelevant ones $\mathcal{P}^- = \mathcal{P} \setminus \mathcal{P}^+$ (as negatives). Therefore, high-quality negatives are critical to the effectiveness of dense retrievers. Existing work commonly leverages the BM25 negatives (Karpukhin et al., 2020) or the top-$k$ ranked negatives mined by a well-trained dense retriever (Qu et al., 2021; Xiong et al., 2021), denoted as $\tilde{\mathcal{D}}^-$. Then, the optimization objective can be formulated as:

$$\theta^* = \arg\min_{\theta} \sum_q \sum_{d^+ \in \mathcal{D}^+} \sum_{d^- \in \tilde{\mathcal{D}}^-} l(s(q, d^+), s(q, d^-)), \tag{2}$$

where $l(\cdot)$ is the loss function. Besides, as the top-$k$ hard negatives may contain false negatives, recent studies (Qu et al., 2021; Ren et al., 2021b; Lu et al., 2022) have adopted knowledge distillation strategies to solve it. They rely on pre-learned cross-encoder rerankers to produce soft labels on $\tilde{\mathcal{D}}^-$, and minimize the KL divergence between the dual encoders' outputs and the soft labels.

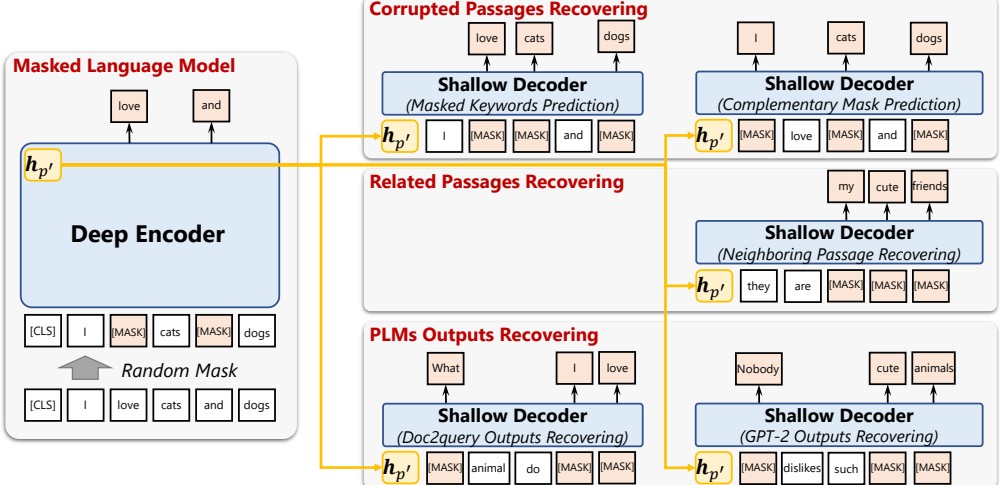

Figure 1: The overview of our MASTER. We incorporate a bottlenecked multi-decoder architecture, and design three types of pre-training tasks, requiring totally five decoders for specific tasks.

## 4 APPROACH

In this section, we present MASTER, an approach to pre-train an effective dense retriever. We first introduce the bottlenecked model architecture (consisting of a deep encoder and multiple shallow decoders), then describe our adopted three types of pre-training tasks unified as the bottlenecked masked autoencoding manner. Figure 1 shows the overview of our approach.

### 4.1 BOTTLENECKED MULTI-DECODER ARCHITECTURE

To pre-train the dense retriever for compressing useful information into the dense vectors, we borrow the idea of bottlenecked masked autoencoders (BMAE) using an encoder-decoder architecture (Liu & Shao, 2022; Wang et al., 2022; Wu et al., 2022). In this way, a deep Transformer encoder is to compress the input text into a dense vector, and a shallow decoder is to recover the input or other related text based on it. To enrich the informativeness of the output dense vectors, we devise a multi-decoder architecture that incorporates five decoders corresponding to different pre-training tasks to capture diverse semantics and relations.

Concretely, the deep Transformer encoder shares the same architecture as BERT (Devlin et al., 2019), and can be initialized with its pre-trained parameters. Given a passage $p$ from the pre-training corpus, we leverage the deep encoder to encode it, and select the output representation of the [CLS] token as its dense vector $\mathbf{h}_p$. Following existing work (Gao & Callan, 2021a; Lu et al., 2021), we employ a masked language model task to pre-train the encoder. Formally, a certain percentage $\alpha\%$ of tokens from $p$ will be masked to obtain $p'$, and the encoder needs to predict them as:

$$L_{\text{MLM}} = \sum_{t_i \in \mathcal{M}_{p'}} - \log p(t_i|p'; \Theta_E) \tag{3}$$

where $\mathcal{M}_{p'}$ denotes the masked tokens in $p'$, $\Theta_E$ denotes the parameters of the encoder. The multiple shallow decoders all adopt the 2-layer bi-directional Transformer encoder architecture, and share the embedding matrix and language modeling head with the deep encoder. For each decoder, its input is an aggressive masked text $x'$ (masking rate $\beta \geq 50\%$) that requires to be recovered. Besides, the dense vector $\mathbf{h}_{p'}$ from the encoder will be inserted into the decoder to replace the original [CLS] token embedding. In this way, the learning objective of each decoder is formulated as:

$$L_D = \sum_{t_i \in \mathcal{M}_{x'}} - \log p(t_i|x', \mathbf{h}_{p'}; \Theta_E, \Theta_D) \tag{4}$$

where $\mathcal{M}_{x'}$ denotes the masked tokens in $x'$, $\Theta_D$ denotes the parameters of the decoder. Such a way can build the information bottleneck that the multiple decoders must rely on the $\mathbf{h}_{p'}$ to help recover its input text, forcing $\mathbf{h}_{p'}$ to reserve useful information from the input.

## 4.2 MULTI-TASK PRE-TRAINING

Based on the architecture, we devise multiple pre-training tasks, to help the output dense vectors from the encoder capture more useful information. Concretely, we adopt three types of tasks to capture the semantic information within passages, relations with other passages, and knowledge from other PLMs, namely corrupted passages recovering, related passages recovering and PLMs outputs recovering, respectively.

**Corrupted Passages Recovering.** Given a passage $p$ from the pre-training corpus, the corrupted passages recovering tasks (CPR) first mask its contained tokens to compose the inputs of the encoder $p'$ and decoder $\hat{p}'$ according to the mask rates $\alpha\%$ and $\beta\%$ respectively. Then, the output dense vector $\mathbf{h}_{p'}$ from the encoder will be leveraged to help the shallow decoder to recover $\hat{p}'$ into $p$. Such a way is helpful to compress important semantic information from the passage into the dense vector. To achieve it, we design two pre-training tasks by utilizing special masking mechanisms for the decoder, namely masked keywords prediction (MKP) and complementary mask prediction (CMP).

For MKP, we aim to mask as more keywords as possible in the decoder, as they may reflect important semantic information of the passage. Specifically, we rely on the widely-used TF-IDF weights (Ramos et al., 2003) to obtain a masked probability distribution about words in the passage, where keywords with low frequencies would receive larger probabilities to be masked. In this way, the input masked passage $\hat{p}'_{\text{MKP}}$ of the decoder will lose most keywords, which will force the dense vector $\mathbf{h}_{p'}$ to well reserve their information for recovering.

For CMP, given the passage $p$, we leverage a complementary mask mechanism in the decoder that masks the unmasked tokens from the input of the encoder $p'$. As a result, the incomplete inputs of the encoder and decoder will be complementary, and the dense vector $\mathbf{h}_{p'}$ should accurately remember all the unmasked input information from $p'$ for recovering $\hat{p}'_{\text{CMP}}$.

Finally, the pre-training objective of the CPR tasks is given by combining the above two tasks as:

$$L_{\text{CPR}} = \sum_{t_i \in \mathcal{M}_{\text{MKP}}} -\log p(t_i | \hat{p}'_{\text{MKP}}, \mathbf{h}_{p'}; \Theta_E, \Theta_D^{\text{MKP}}) + \sum_{t_i \in \mathcal{M}_{\text{CMP}}} -\log p(t_i | \hat{p}'_{\text{CMP}}, \mathbf{h}_{p'}; \Theta_E, \Theta_D^{\text{CMP}}), \quad (5)$$

where $\mathcal{M}_{\text{MKP}}$ and $\mathcal{M}_{\text{CMP}}$ denote the masked tokens in $\hat{p}'_{\text{MKP}}$ and $\hat{p}'_{\text{CMP}}$, respectively, and $\Theta_D^{\text{MKP}}$ and $\Theta_D^{\text{CMP}}$ are the parameters of the two specific decoders, respectively.

**Related Passages Recovering.** The related passages recovering task (RPR) aims to model the semantic relationships between related passages. In this work, we focus on the commonly-used and easily-obtained co-occurrence relation from the pre-training corpus. Based on this motivation, we collect the passage pairs $\{\langle p_i, p_{i+1} \rangle\}$ that are neighbouring spans in a document, and devise the neighbouring passage recovering task (NPR).

In NPR, given a neighbouring passage pair $\langle p_i, p_{i+1} \rangle$, we rely on the mask rates $\alpha\%$ and $\beta\%$ to mask their tokens for composing the inputs of the encoder $p'_i$ and decoder $p'_{i+1}$, respectively. Next, the output dense vector of $p'_i$ from the deep encoder is utilized to help the decoder recover $p'_{i+1}$. Such a way will encourage the dense vector to retain the information that is useful to recover the neighbouring passage, capturing the intrinsic token-level correlations across the two passages. Besides, we also rely on the TF-IDF weights of words to mask more keywords in the decoder as MKP, which further increases the difficulty of this task and forces the dense vector to focus more on the key information. The learning objective of the RPR task can be defined as:

$$L_{\text{RPR}} = \sum_{t_i \in \mathcal{M}_{\text{NPR}}} -\log p(t_i | p'_{i+1}, \mathbf{h}_{p'_i}; \Theta_E, \Theta_D^{\text{NPR}}), \quad (6)$$

where $\mathcal{M}_{\text{NPR}}$ and $\Theta_D^{\text{NPR}}$ denote the masked tokens in $p'_{i+1}$ and the parameters of the decoder specially for the NPR task, respectively. Note that existing work (Lee et al., 2019; Ma et al., 2022) has also considered the neighbouring relations and mostly adopts the contrastive learning objective to capture it. In fact, contrastive learning mainly aims to characterize the passage-level semantics and arbitrarily pushes apart irrelevant passages, even if they are semantically relevant. As a comparison, the NPR task can capture more fine-grained token-level characteristics, and such a text covering task is much safer to not hurt the semantic relevance between relevant but not neighbouring passages.

**PLMs Outputs Recovering.** The above tasks are able to capture the semantic information and relations within the unsupervised pre-training corpus. We further consider to learn the knowledge from other PLMs, to capture more rich information beyond the corpus. Based on this idea, we design the PLMs outputs recovering tasks (POR) that aim to recover the outputs of two generative PLMs, consisting of the doc2query outputs recovering (DOR) and GPT-2 outputs recovering (GOR) tasks.

Given a passage $p$, we leverage a public well-trained doc2query model (Nogueira & Lin, 2019) to generate $k$ relevant queries $\{q_i\}_{i=1}^k$ and concatenate them into a long sentence $s_{(q)}$, as the generated queries have shown effectiveness in previous dense retrieval methods (Nogueira et al., 2019). Besides, we also use $p$ as the prompt to guide the popular autoregressive GPT-2 model (Radford et al., 2019) to generate a long sentence $s_{(g)}$, as GPT-2 has shown surprising performance in generating informative long text. Then, we aggressively mask the tokens in $s_{(q)}$ and $s_{(g)}$ according to the mask rate $\beta\%$, to obtain the inputs $s'_{(q)}$ and $s'_{(g)}$ of two task-specific decoders. Similar to above tasks, the two decoders also rely on the dense vector $\mathbf{h}_{p'}$ to recover the generated texts, and the pre-training objective of the POR tasks is the combination of the two tasks as:

$$L_{\text{POR}} = \sum_{t_i \in \mathcal{M}_{\text{DOR}}} -\log p(t_i|s'_{(q)}, \mathbf{h}_{p'}; \Theta_E, \Theta_D^{\text{DOR}}) + \sum_{t_i \in \mathcal{M}_{\text{GOR}}} -\log p(t_i|s'_{(g)}, \mathbf{h}_{p'}; \Theta_E, \Theta_D^{\text{GOR}}), \quad (7)$$

where $\mathcal{M}_{\text{DOR}}$ and $\mathcal{M}_{\text{GOR}}$ denote the masked tokens in $s'_{(q)}$ and $s'_{(g)}$, respectively, and $\Theta_D^{\text{DOR}}$ and $\Theta_D^{\text{GOR}}$ are the parameters of the two specific decoders, respectively. In this way, we can enhance the dense vector to capture richer semantics from other PLMs, and learn more information not included in the documents corpus. Note that such a way is similar to the knowledge distillation process that transfers the learned knowledge from PLMs into the dense vector by forcing it to predict the PLMs' outputs.

### 4.3 LEARNING

During pre-training, we optimize the parameters in the deep encoder and the multiple shallow decoders using the above pre-training tasks, denoted as:

$$L_{\text{total}} = L_{\text{MLM}} + L_{\text{CPR}} + L_{\text{RPR}} + L_{\text{POR}} \quad (8)$$

During fine-tuning, we utilize the pre-trained deep encoder as the backbone of the query and passage encoders. Following the pipeline in previous dense retrieval methods (Gao & Callan, 2022; Wang et al., 2022; Wu et al., 2022), we first train the **Retriever**$_1$ using the in-batch negatives and BM25 hard negatives. Then, we utilize Retriever$_1$ to mine hard negatives from a large-scale passage pool, and leverage these negatives and in-batch negatives to train the **Retriever**$_2$. Next, we train a cross-encoder reranker model based on the mined negatives from Retriever$_2$. Finally, we distil the knowledge from the reranker into the **Retriever**$_{\text{distil}}$ by using it to produce soft labels for both positives and mined negatives from Retriever$_2$. Note that our pre-trained encoder is used to initialize the Retriever$_1$, Retriever$_2$ and Retriever$_{\text{distil}}$.

## 5 EXPERIMENT

### 5.1 EXPERIMENTAL SETTING

**Datasets and Evaluation.** We conduct experiments on several text retrieval datasets: MS MARCO Passage Ranking (MS-MARCO) (Nguyen et al., 2016), TREC 2019 Deep Learning Track (TREC-19) (Craswell et al., 2020), TREC 2020 Deep Learning Track (TREC-20) (Craswell et al., 2021), and Natural Questions (NQ) (Kwiatkowski et al., 2019). The statistics of the above datasets are shown in Table 1. MS-MARCO consists of real queries collected from Bing search engine. NQ is an open domain question answering dataset.

Table 1: Statistics of the text retrieval datasets. "#Passage" means the number of passages.

| Dataset | Train | Dev | Test | #Passage |
|---|---|---|---|---|
| MS-PAS | 502,939 | 6,980 | - | 8,841,823 |
| TREC-19 | - | - | 200 | 8,841,823 |
| TREC-20 | - | - | 200 | 8,841,823 |
| NQ | 58,880 | 8,757 | 3,610 | 21,015,324 |
| TQ | 60,413 | 8,837 | 11,313 | 21,015,324 |

Table 2: Experimental results on three web search datasets. The best and second-best methods are marked in bold and underlined, respectively. The ✓in the column of "with KD?" means that the model has used knowledge distillation.

| Model | with KD? | MS-MARCO | | | TREC-19 | TREC-20 |
| | | MRR@10 | R@50 | R@1k | nDCG@10 | nDCG@10 |
| --- | --- | --- | --- | --- | --- | --- |
| BM25 (Yang et al., 2017) | | 18.5 | 58.5 | 85.7 | 51.2 | 47.7 |
| DeepCT (Dai & Callan, 2019a) | | 24.3 | 69.0 | 91.0 | 57.2 | - |
| docT5query (Nogueira & Lin, 2019) | | 27.7 | 75.6 | 94.7 | 64.2 | - |
| ANCE (Xiong et al., 2021) | | 33.0 | - | 95.9 | 64.5 | 64.6 |
| STAR (Zhan et al., 2021) | | 34.7 | - | - | 68.3 | - |
| TAS-B (Hofstätter et al., 2021) | ✓ | 34.0 | - | 97.5 | 71.2 | 69.3 |
| RocketQA (Qu et al., 2021) | ✓ | 37.0 | 85.5 | 97.9 | - | - |
| RocketQAv2 (Ren et al., 2021b) | ✓ | 38.8 | 86.2 | 98.1 | - | - |
| AR2 (Zhang et al., 2022) | ✓ | 39.5 | 87.8 | 98.6 | - | - |
| ERNIE-Search (Lu et al., 2022) | ✓ | 40.1 | 87.7 | 98.2 | - | - |
| COIL (Gao et al., 2021a) | | 35.5 | - | 96.3 | 70.4 | - |
| ColBERT (Khattab & Zaharia, 2020) | | 36.0 | 82.9 | 96.8 | - | - |
| ColBERTv2 (Santhanam et al., 2022) | ✓ | 39.7 | 86.8 | 98.4 | - | - |
| SEED (Lu et al., 2021) | | 33.9 | - | 96.1 | - | - |
| RetroMAE (Liu & Shao, 2022) | | 35.0 | - | 97.6 | - | - |
| Condenser (Gao & Callan, 2021a) | | 36.6 | - | 97.4 | 69.8 | - |
| coCondenser (Gao & Callan, 2022) | | 38.2 | 86.5 | 98.4 | 71.7 | 68.4 |
| CoT-MAE (Wu et al., 2022) | | 39.4 | 87.0 | 98.7 | - | 70.4 |
| PAIR (Ren et al., 2021a) | ✓ | 37.9 | 86.4 | 98.2 | - | - |
| SimLM (Wang et al., 2022) | ✓ | 41.1 | 87.8 | 98.7 | 71.2 | 69.7 |
| MASTER | ✓ | **41.5** | **88.6** | **98.8** | **72.7** | **71.7** |

## 5.2 MAIN RESULTS

**Performance on Web Search Datasets.** Table 2 shows the experimental results on three web search benchmarks, *i.e.,* MS-MARCO, TREC-2019 and TREC-2020. First, we can see that with or without distillation strategy, the best baselines are both pre-training dense retrieval methods, *i.e.,* CoT-MAE and SimLM, even outperforming methods using multiple representations. It indicates that proper pre-training strategies are helpful to the downstream dense passage retrieval tasks. Second, SimLM mostly outperforms other baselines. It employs a bottlenecked architecture that learns to compress the input information into a dense vector, and adopts a replaced language modeling objective to pre-train it. Such a way is more effective to force the dense vector to reserve the important semantics.

Besides, we can see that our approach outperforms all the baselines in terms of all metrics on all datasets. Our approach adopts a multi-task pre-training framework that unifies five tasks on recovering of corrupted passages, related passages and PLMs outputs, based on a bottlenecked one-encoder multi-decoder architecture. In this way, we can force the output dense vector from the encoder to be more informative and functional to accomplish these tasks, leading to better representative capacity.

**Performance on Open Domain QA Datasets.**
Table 3 shows the experimental results an open domain QA datasets, NQ. For a fair comparison, we only report the performance of Retriever$_2$ without performing knowledge distillation in our approach. First, we can also see that pre-training dense retrieval methods mostly outperform other methods. It further indicates the effectiveness of pre-training techniques in open domain QA tasks. Besides, coCondenser and SimLM perform better than other methods, the reason is that they both adopt a bottlenecked architecture to compress the information into the dense vectors. Finally, we can see that our approach outperforms all the baselines. As a comparison, our approach can enhance the informativeness of dense vectors by integrating

Table 3: Results on NQ. Following (Wang et al., 2022), we report the performance of Retriever$_2$ without knowledge distillation for our MSATER.

| Model | NQ | |
| | R@20 | R@100 |
| --- | --- | --- |
| BM25 (Yang et al., 2017) | 59.1 | 73.7 |
| DPR$_{single}$ (Karpukhin et al., 2020) | 78.4 | 85.4 |
| ANCE (Xiong et al., 2021) | 81.9 | 87.5 |
| RocketQA (Qu et al., 2021) | 82.7 | 88.5 |
| RocketQAv2 (Ren et al., 2021b) | 83.7 | 89.0 |
| Condenser (Gao & Callan, 2021a) | 83.2 | 88.4 |
| PAIR (Ren et al., 2021a) | 83.5 | 89.1 |
| coCondenser (Gao & Callan, 2022) | 84.3 | 89.0 |
| SimLM (Wang et al., 2022) | 84.3 | 89.3 |
| MASTER | **84.6** | **89.4** |

Table 4: Zero-shot dense retrieval nDCG@10 performances on BEIR benchmark. Results with * are from our reproduction.

| Dataset | BERT | LaPraDoR | SimCSE | DiffCSE | SEED | Condenser | SimLM* | MASTER |
|---|---|---|---|---|---|---|---|---|
| TREC-COVID | 0.649 | 0.495 | 0.524 | 0.492 | 0.612 | **0.754** | 0.637 | 0.620 |
| BioASQ | 0.262 | 0.239 | 0.264 | 0.258 | 0.297 | 0.317 | 0.350 | **0.354** |
| NFCorpus | 0.257 | 0.283 | 0.250 | 0.259 | 0.256 | 0.278 | 0.323 | **0.330** |
| NQ | 0.438 | 0.415 | 0.412 | 0.412 | 0.425 | 0.459 | 0.477 | **0.516** |
| HotpotQA | 0.478 | 0.488 | 0.502 | 0.499 | 0.528 | 0.537 | 0.581 | **0.589** |
| FiQA-2018 | 0.237 | 0.266 | 0.240 | 0.229 | 0.244 | 0.261 | 0.292 | **0.328** |
| Signal-1M (RT) | 0.216 | 0.245 | **0.264** | 0.260 | 0.246 | 0.258 | 0.257 | 0.252 |
| TREC-NEWS | 0.362 | 0.206 | 0.368 | 0.363 | 0.335 | 0.353 | 0.326 | **0.409** |
| Robust04 | 0.364 | 0.310 | 0.353 | 0.343 | 0.348 | 0.352 | 0.368 | **0.405** |
| ArguAna | 0.357 | **0.503** | 0.436 | 0.468 | 0.347 | 0.375 | 0.421 | 0.395 |
| Touché-2020 | 0.270 | 0.178 | 0.178 | 0.168 | 0.180 | 0.223 | 0.292 | **0.320** |
| CQADupStack | 0.284 | 0.326 | 0.295 | 0.305 | 0.285 | 0.316 | **0.332** | 0.327 |
| Quora | 0.782 | 0.843 | 0.848 | 0.850 | 0.849 | **0.855** | 0.773 | 0.791 |
| DBPedia | 0.298 | 0.328 | 0.304 | 0.303 | 0.324 | 0.331 | 0.345 | **0.399** |
| SCIDOCS | 0.115 | **0.145** | 0.125 | 0.125 | 0.117 | 0.136 | **0.145** | 0.141 |
| FEVER | 0.684 | 0.518 | 0.651 | 0.641 | 0.653 | 0.682 | 0.657 | **0.692** |
| Climate-FEVER | 0.205 | 0.172 | **0.222** | 0.200 | 0.176 | 0.199 | 0.163 | 0.215 |
| SciFact | 0.504 | 0.483 | 0.545 | 0.523 | 0.556 | 0.570 | 0.588 | **0.637** |
| **Avg. Performance** | 0.376 | 0.358 | 0.377 | 0.372 | 0.377 | 0.403 | 0.407 | **0.429** |

Table 5: Comparison with different pre-training dense retrieval methods in three stages of our fine-tuning pipeline on the dev set of MS-MARCO. Results with * are from our reproduction.

| Model | coCondenser | | CoTMAE | | SimLM | | MASTER | |
|---|---|---|---|---|---|---|---|---|
| | MRR@10 | R@1k | MRR@10 | R@1k | MRR@10 | R@1k | MRR@10 | R@1k |
| Retriever$_1$ | 35.7 | 97.8 | 36.8* | 98.3* | 38.0 | 98.3 | **38.3** | **98.8** |
| Retriever$_2$ | 38.2 | 98.4 | 39.2 | 98.7 | 39.1 | 98.6 | **40.4** | **98.8** |
| Retriever$_{distil}$ | 40.2 | 98.3 | 40.4 | 98.7 | 41.1 | 98.7 | **41.5** | **98.8** |

multiple pre-training tasks, which compress the semantic information within passages, model the relations between passages, and learn the knowledge from other PLMs.

**Zero-Shot Evaluation.** We evaluate the zero-shot retrieval performance of our approach on BEIR benchmark (Thakur et al., 2021). It contains 18 datasets, covering dense retrieval tasks across different domains, *e.g.,* question answering, fact checking, bio-medical retrieval and news retrieval. Following Thakur et al. (2021), we fine-tune our approach in MS-MARCO training set and evaluate it on the BEIR benchmark using the official evaluation toolkit [1] to show the zero-shot abilities. nDCG@10 is chosen as the evaluation metrics. As shown in Table 4, we can see that our approach outperforms most of baselines in all 18 datasets, and the average performance also surpasses all baselines significantly. Since our approach incorporates multiple pre-training tasks for learning the dense representations, such a way can enrich the informativeness of them and help better adapt into different domains and retrieval tasks.

## 5.3 FURTHER ANALYSIS

**Performance in Three Stages of Fine-tuning Pipeline.** To further investigate the effectiveness of our approach, we show the performances of MASTER and other pre-training dense retrieval methods in each stage of our fine-tuning pipeline. Here, the models in the three stages are all initialized by corresponding pre-trained parameters of these methods. As shown in Table 5, we can see that the performances of all pre-training methods are consistently improving with the process of the three-stage training, and the models in the third stage Retriever$_{distil}$ perform the best. In addition, our approach also consistently outperforms all other pre-training methods in all the three stages. It also indicates the effectiveness of our proposed multi-task pre-training strategy, which can produce higher-quality dense vectors for downstream retrieval tasks.

**Ablation and Variation Study.** Our proposed approach incorporates a multi-decoder architecture and three types of tasks for pre-training. To verify the effectiveness of each part, we conduct the

[1] https://github.com/beir-cellar/beir

Table 6: Ablation and variation study of our approach. We report the means and variances of MRR@10 of the retriever$_1$ and retriever$_2$ on the dev set of MS-MARCO.

| Model | MASTER | w/o CPR | w/o RPR | w/o POR | +Shared-Dec | SimLM |
|---|---|---|---|---|---|---|
| Retriever$_1$ | **38.3** (0.016) | 37.7 (0.016) | 37.6 (0.007) | 37.6 (0.016) | 37.4 (0.016) | 38.0 |
| Retriever$_2$ | **40.4** (0.016) | 39.9 (0.007) | 39.8 (0.009) | 39.8 (0.002) | 39.1 (0.069) | 39.1 |

Table 7: Performance comparison w.r.t. different training data size on the dev set of MS-MARCO.

| Model | MRR@10 | | | | | R@1k | | | | |
|---|---|---|---|---|---|---|---|---|---|---|
| | 5% | 10% | 20% | 50% | 100% | 5% | 10% | 20% | 50% | 100% |
| SimLM | 31.5 | 32.6 | 33.8 | 35.4 | 38.0 | 97.2 | 97.5 | 97.8 | 98.2 | 98.7 |
| MASTER | 33.0 | 34.3 | 35.4 | 36.6 | **38.3** | 97.8 | 97.9 | 98.4 | 98.7 | **98.8** |

ablation and variation study on the dev set of MS-MARCO to analyze their contributions. We remove the CPR, RPR and POR tasks individually, and propose a variants that adopts a shared decoder to deal with the multiple tasks. As shown in Table 6, we can see that all the ablation and variation models will lead to the performance degradation. It indicates that all the pre-training tasks and our multi-decoder architecture are useful to improve the performance. Besides, after removing any type of pre-training tasks, our Retriever$_2$ still outperforms the SOTA method, SimLM. It further shows the promising effectiveness of multi-task pre-training for dense retrieval tasks.

**Performance w.r.t. Different Pre-training Steps.** As a pre-training approach, the number of pre-training steps will affect the performance on downstream tasks. In each step, we optimize the model parameters using a batch of pre-training data by gradient descent algorithm. However, too many pre-training steps are time-consuming and costly. Here, we investigate the performance convergence speed of our approach during pre-training. As shown in Figure 2, we can see that our model performs well with few pre-training steps, especially that the retriever$_2$ of our method achieves the 39.1 on MRR@10 metric (the same as SimLM) after 10k steps. It shows that our approach is more effective to pre-train effective dense vectors, with no need for too many pre-training steps.

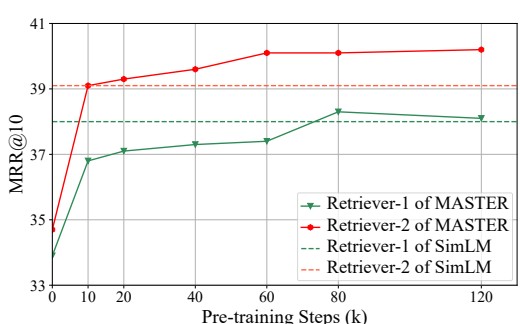

Figure 2: Performance comparison w.r.t. different number of pre-training steps on MS-MARCO.

**Few-Shot Learning.** In our approach, as we have pre-trained the backbone via a multi-task manner, the pre-trained dense vectors can be easily adapted into downstream tasks with less data. To validate this conjecture, we reduce the training data size into 50%, 20%, 10% and 5%, and compare the MRR@10 and R@1k of our approach with the SOTA pre-training method SimLM. As shown in Table 7, we can see that the performance substantially drops when less training data is used. Additionally, our approach is consistently better than SimLM in all cases, especially in an extreme sparsity level (5%). It indicates that MASTER is better pre-trained to effectively adapt to downstream dense retrieval task.

## 6 CONCLUSION

In this paper, we proposed MASTER, a multi-task pre-trained bottlenecked masked autoencoder for dense retrieval task. In our approach, we adopted a bottlenecked multi-decoder architecture to integrate a variety of pre-training tasks, and devised three types of pre-training tasks about corrupted passages recovering, related passage recovering and PLMs outputs recovering. The three types of tasks focused on compressing the semantic information within the passages, modeling relations among passages, and learning the knowledge from external public generative PLMs, respectively, leading to more informative and effective dense vectors. Experimental results have shown that our approach outperforms several competitive baselines.

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

Table 8: Experimental results on four NLU tasks from GLUE.

| Model | CoLA | MRPC | STS-B | QQP |
|-------|------|------|-------|-----|
| BERT | 59.1 | 87.7 | 87.8 | 89.7 |
| Ours | 60.7 | 89.1 | 88.0 | 89.8 |

## A  BASELINES.

We compare our approach with a variety of methods:

• **BM25** (Yang et al., 2017) is a widely-used sparse retriever based on exact matching. **DeepCT** (Dai & Callan, 2019b) and **docT5query** (Nogueira & Lin, 2019) enhance BM25 with neural models.

• **ANCE** (Xiong et al., 2021), **TAS-B** (Hofstätter et al., 2021) and **STAR** (Zhan et al., 2021) are dense retrieval methods that adopt top-$k$ hard negatives to improve training. **RocketQA** (Qu et al., 2021), **AR2** (Zhang et al., 2022) and **ERNIE-search** (Lu et al., 2022) utilize knowledge distillation technique that leverages a teacher model to guide the training of the dual-encoder retriever.

• **COIL** (Gao et al., 2021b), **ColBERT** (Khattab & Zaharia, 2020) and **ColBERTv2** (Santhanam et al., 2022) utilize multiple representations for text retrieval.

• **SEED** (Lu et al., 2021), **RetroMAE** (Liu & Shao, 2022), **Condenser** (Gao & Callan, 2021b), **PAIR** (Ren et al., 2021a), **coCondenser** (Gao & Callan, 2022), **CoT-MAE** (Wu et al., 2022) and **SimLM** (Wang et al., 2022) design special pre-training tasks to improve the backbone models.

## B  IMPLEMENTATION DETAILS.

During pre-training, we leverage BERT-base to initialize the shared encoder, and the multiple decoders are randomly initialized two-layer Transformers. Following previous work (Gao & Callan, 2022; Wu et al., 2022; Wang et al., 2022), we leverage the passages in MS-MARCO and NQ dataset as the pre-training corpus of them, respectively. The pre-training steps are setting to 120k. During fine-tuning, we also follow SimLM that first train $Retriever_1$ using BM25 negatives, then train $Retriever_2$ using the hard negatives mined by $Retriever_1$, finally utilize a cross-encoder based reranker [2] to perform knowledge distillation on the hard negatives mined by $Retriever_2$, to train $Retriever_{distil}$. Note that our pre-trained deep Transformer encoder is leveraged to initialize the parameters of $Retriever_1$, $Retriever_2$ and $Retriever_{distil}$. Our all other hyper-parameters are the same as SimLM (Wang et al., 2022). All the experiments in this work are conducted on 8 NVIDIA Tesla A100 GPUs.

## C  NATURAL LANGUAGE UNDERSTANDING TASKS

In our approach, as we have integrated multiple pre-training tasks for learning, our model is able to capture diverse knowledge from these tasks. In this part, we aim to evaluate if our pre-training methods can also benefit for other tasks, *i.e.,* natural language understanding (NLU). We select the single-sentence and similarity tasks from the GLUE benchmark (Wang et al., 2019) (*i.e.,* CoLA, MRPC, STS-B and QQP), which focus on predicting the acceptability, similarity and paraphrase of sentences from different domains (e.g., news and misc). We fine-tune our pre-trained model on these tasks. and all the hyper-parameters are following the suggestions of the original BERT paper (Devlin et al., 2019). As shown in Table 8, we can see that our approach can also improve the performance of BERT on these NLU tasks. It indicates that our multi-task pre-training can also enrich the useful knowledge about NLU tasks for the PLM.

## D  HYPER-PARAMETER TUNING.

Our approach has two important hyper-parameters, namely the masked rates of the deep encoder and multiple decoders, as they control the information bottleneck of our approach. In this part, we

---

[2]Following SimLM, we also leverage ELECTRA-base to initialize the reranker and train it using the mined negatives by $Retriever_2$.

Table 9: Performance comparison w.r.t. different masked rates in the encoder and decoder. We report MRR@10 of the $Retriever_1$ and $Retriever_2$ on the dev set of MS-MARCO.

| Model | 30% En-50% De | 15% En-50% De | 50% En-50% De | 30% En-30% De | 30% En-70% De |
|---|---|---|---|---|---|
| $Retriever_1$ | **38.3** | 37.9 | 37.6 | 37.5 | 38.0 |
| $Retriever_2$ | **40.4** | 39.9 | 39.7 | 39.8 | 39.9 |

set the masked rate in the encoder to be 15%, 30% and 50%, and that in decoders to be 30%, 50% and 70%. Table 9 shows the evaluation results. First, we can see that our model is robust to these different hyper-parameter settings. Besides, when the masked rates of the encoder and decoders are set to 30% and 50% respectively, our model performs slightly better than others. Therefore, we apply 30% and 50% as the masked rates of the encoder and decoders.

