# OpenReview forum: "MASTER: Multi-task Pre-trained Bottlenecked Masked Autoencoders are Better Dense Retrievers"
_ICLR.cc/2023/Conference — Submitted to ICLR 2023_

### Official Review · Reviewer_3P7A · 2022-10-25

**Confidence:** 5
**Correctness:** 3
**Technical Novelty And Significance:** 2
**Empirical Novelty And Significance:** 3
**Recommendation:** 6

**Clarity, Quality, Novelty And Reproducibility:**

The paper is clear although some sentences are quite obscure (especially in section 4.2 where the loss is described). The paper seems to be reproducible (open source code would be a plus, but is not provided). Novelty-wise, the losses are (somehow) novel to the extent of my knowledge, being somehow inspired by Electra as acknowledged by the authors.


Some points:
- Results reported in table 5 show that losses are quite redundant: it would have been good to do ablations with just one loss to see the effect of each (and also to run experiments several time to get an idea of the variance of the results).

Minor points:
- p.5: "and remains the masked ones unchanged"? "that can deduce its neighbouring relations with other passages."?
- Figure 1: the "complementary mask prediction" figure part is wrong
- Table 7: which dataset?
- Figure 2: what is a step is not defined



**Strength And Weaknesses:**

Strengths:

- Unifying/studying the pre-training of information retrieval
- Improved performances with respect to the state-of-the-art.
- The proposed pre-training is simple to set up, and might be a basis for future pre-training

Weaknesses:

- it is not obvious why the fact that using a more homogeneous task is better: There is a bold statement p.2: "due to the divergences of input formats and learning objectives among different tasks, an arbitrary integration of these tasks is inappropriate, which may even cause detrimental gradient interference, leading to performance degradation" - how is your setting more appropriate? Why having multiple tasks be detrimental - I checked the references and they do not provide any reason (but it is true that the magnitude of the respective losses has to be set as in ref. Kendall et al. 2018).
- no comparison on the BEIR dataset (which would show how robust the pre-training + training procedure is)
- apart from an ablation, there is no further analysis of the 5 proposed losses, although they are combined directly with no discussion on why they should have the same importance.
- No code is provided
- Using DocT5 (POR loss) shows very moderate improvement in comparison to the cost: it relies on a query generation model (trained on MS Marco which is used for the experiments here...) and document generation (which is costly). I fail to see why it justifies the 0.4 drop in MRR@10 (especially since the variance of the results is not really known)


**Summary Of The Paper:**

This paper proposes to pre-train a dense retriever with multiple tasks (five in total), which are all related to masked language modelling (multi-decoder). Results on MS-Marco show moderate improvement with respect to baselines.


**Summary Of The Review:**

This paper proposes a new pre-training for IR - even if the authors frame it as "multi-task", it is still heavily based on a masked language modelling. Results are good but not so impressive when looking at the SimLM baseline (which shares this idea of corruption loss) and especially at the quite expensive pre-training procedure.

---

> ### Author Response · Authors · 2022-11-19
> **Response to the Concerns of the Reviewer**
>
> Thanks for your insightful suggestions and we have listed our response to your concerns as follows. If you also have any other questions, please feel free to let us know. We will continue to try our best to answer for you.
>
> 1. **For the question about using a more homogeneous task is better:** Empirically, we have conducted experiments by simply combining several commonly-used pre-training tasks (contrastive learning like Cocondenser and bottleneck MLM like SimLM) and see that the model does not perform well (Retriever-1: 36.6). Compared with SimLM that only adopts the bottleneck MLM (Retriever-1: 38.0), we can conclude that the performance actually degrades a lot and an arbitrary integration of these tasks is inappropriate. In our manuscript, the cited papers (Kendall et al., 2018; Yu et al., 2020) have pointed out the possible hidden problems of such results, e.g., gradient interference.
>
> 2. **For the question about the BEIR dataset:** Following the reviewer's suggestion, we have conducted experiments on the BEIR dataset. We report the results in Table 4 of our new manuscript, and we summarize the average performance between our approach and several baselines in the following table. From the table, we can see that the average performance of our approach can outperform all baselines.  Besides, as shown in Table 4 of our manuscript, our approach surpasses all baselines in most settings. All these results indicate the effectiveness and robustness of our approach in zero-shot setting.
>
> \\begin{array} {|l|c|c|c|c|c|c|c|c|}
> \\hline
> \\textrm{Method} & \\textrm{BERT} & \\textrm{LaPraDoR} & \\textrm{SimCSE} & \\textrm{DiffCSE} & \\textrm{SEED} & \\textrm{Condenser} & \\textrm{SimLM} & \\textrm{Ours} \\\\
> \\hline
> \\textrm{Avg. Performance} & 0.376 & 0.358 & 0.377 & 0.372 & 0.377 & 0.403 & 0.407 & 0.429  \\\\
> \\hline
> \\end{array}
>
> 3. **For the question about further analysis of five losses**: we conduct a more detailed ablation study here to show the importance of each loss in our approach. We report the MRR@10 performance of Retriever-1 and Retriever-2 on MS-MARCO-Passage dataset ain the following table. We can see that all these variations perform not better than our approach (38.3, 40.4). It indicates that all these pre-training objectives are useful for improving the model performance. Indeed, we assign the same importance to all losses just for convenience. Inspired by the suggestions of this reviewer, we conduct an exploratory experiment that gives dynamic weights for all losses w.r.t. their values to guarantee that all losses are of the same magnitude. We can see that such an easy way can also improve the performance of our MASTER (from 38.3, 40.4 to 38.6, 40.6). We consider further investigating this direction and releasing a more effective pre-trained model for the dense retrieval task.
>
> \\begin{array} {|l|c|c|c|c|c|c|}
> \\hline
> \\textrm{MRR}@10 & \\textrm{Ours} & \\textrm{w/o MKP} & \\textrm{w/o CMP} & \\textrm{w/o RPR} & \\textrm{w/o DOR} & \\textrm{w/o GOR} \\\\
> \\hline
> \\textrm{Retriever}_1 & 38.3  & 38.0 & 38.0 & 37.6 & 37.8 & 37.8 \\\\
> \\hline
> \\textrm{Retriever}_2  & 40.4  & 40.0 & 40.0 & 39.9 & 39.8 & 40.0 \\\\
> \\hline
> \\end{array}
>
> 4. **For the reproducibility question**: We have prepared our approach's code and uploaded it to the ICLR rebuttal page. And we will release our pre-trained checkpoints after the review period to guarantee that everyone can re-implement the experimental results in our paper.
>
> 5. **For the question about DocT5**: as shown in General Response, we retrain our approach and the variations three times using different random seeds and report their means and variances. We can see that removing POR loss would lead to the performance mean values degradation a lot. It shows the effectiveness of the POR loss. Besides, the cost of the POR loss is not enormous, as we directly reuse the publicly released DocT5 and GPT-2 checkpoints in huggingface. By pre-generating the relevant queries from DocT5 before training, we do not require to load such a large PLM during the training process. Our experiments have found that adding the POR loss only increases 1/3 the pre-training time, leading to stably better performance in the three-stage fine-tuning pipeline.
>
> 6. **For the obscure clarity problem**: we have modified the mentioned unclear sentences and have carefully read and checked the writing of our manuscript for better clarification of our approach.

---

> > ### Comment · Reviewer_3P7A · 2022-11-25
> > **Thanks for the answers**
> >
> > Given the new experiments conducted on BEIR (and their results), the answers to my questions (and in the updated PDF), and the future availability of the code, I will raise my recommendation to "accept"

---

> > > ### Author Response · Authors · 2022-12-02
> > > **Appreciate your new response**
> > >
> > > **We sincerely thank the reviewer for the updated comments and score.**
> > >
> > > We have prepared our code into an anonymous github Webpage, i.e.,**https://anonymous.4open.science/r/MASTER-FF38/README.md**, and show how to pre-train and fine-tune a dense retrieval model using our provided code. After the reviewer period, we will further make the pre-trained and fine-tuned checkpoints public for the utilization of other researchers. If you also have any other questions, please feel free to let us know. We will continue to try our best to answer for you.

---

> > > > ### Author Response · Authors · 2022-12-05
> > > > **Release our Pre-trained Corpus and Checkpoints**
> > > >
> > > > We have just finished preparing our **pre-training corpus and checkpoints** in the anonymous Webpage, i.e.,**https://anonymous.4open.science/r/MASTER-FF38/README.md**, and provided detailed command lines for using them to implement our pre-training process or fine-tuning. If you have any questions or concerns about them, please feel free to tell us.

---

### Official Review · Reviewer_jK4d · 2022-10-25

**Confidence:** 3
**Correctness:** 3
**Technical Novelty And Significance:** 2
**Empirical Novelty And Significance:** 2
**Recommendation:** 5

**Clarity, Quality, Novelty And Reproducibility:**

The paper is well written, easy to understand and covers a lot of related work. The proposed pretraining tasks are somewhat novel, but are not sufficiently analyzed. The paper puts a lot of emphasis on its method surpassing the previous state-of-the-art. ~Reproducibility is limited, as most model hyperparameters are not explicitly provided, and no code is available.~

**Strength And Weaknesses:**

Strengths:

* simple model that works well
* paper is easy to understand

Weaknesses (**UPDATED AFTER REBUTTAL**):
* the ablation results are not convincing:
    * the improvements from each pretraining objective are relatively small
    * ~could it be that the results are simply better because you effectively train on more examples the more pretraining objectives you have? Figure 2 suggests that convergence might not have been reached, and hence more pretraining objectives would mean that you actually do more training steps (in the sense of training examples) than what the x-axis shows. When removing each of the 5 pretraining objectives, you could control for this by instead training for 1/5 = 20% longer.~
    * there is no qualitative analysis that shows us that the different pretraining objectives actually teach the model distinct "skills". This could be done by training a model with only a single pretraining objective and then doing an error analysis for each.
* ~in your comparisons to SimLM, how do you make sure that the small differences in performance are not actually due to factors of variation other than the pretraining objective? Are all the model details the same between the two models? How do you make sure that both models receive equal treatment in terms of hyperparameter tuning?~
* the model is of limited novelty; neither the bottlenecked model architecture, nor the multi-task pretraining idea are novel. There is some novelty in the concrete pretraining tasks, but their effect is unclear.

**Summary Of The Paper:**

The paper proposes a multi-task pretraining objective for dense retrieval models. By leveraging the bottlenecked masked autoencoder architecture, the encoder produces a single-vector embedding, which is used to condition the decoder models on three different types of pretraining tasks, which are all trained in parallel: recovery of corrupted (input) passages, recovery of related passages, and recovery of the outputs from an external pretrained LM.
The model is tested on standard web search and open domain QA datasets, and achieves state-of-the-art performance.

**Summary Of The Review:**

The paper proposes a simple method that reaches state-of-the-art in dense retrieval. However, the ablations are not sufficient to convincingly show that the modeling contributions are actually responsible for this success. Hence, the paper would need a major revision before it can be accepted.

---

> ### Author Response · Authors · 2022-11-19
> **Response to the Concerns about the Ablation Results:**
>
> Thanks for your insightful suggestions and we have listed our response to the concerns about the ablation results as follows. If you still have any other questions, please do not hesitate to tell us. We will continue to try our best to answer for you.
>
> 1.**For the question about the ablation results are not convincing**:
>
> **A. Actually, all the pre-training objectives contribute to the improvement of our approach.** To better show the effectiveness of each objective, we follow the reviewer's suggestion, fine-tune the ablation variations and our approach three times and report their means and variances. Table 2 in the General Response shows the results. We can see that removing each pre-training objective would lead to performance degradation, and their low variances (<0.07) indicate that each objective stably improves the model performance.
>
> **B. As shown in the General Response, we have considered this issue and tried the checkpoints using more pre-training steps for all the ablation variations (e.g., 120k, 160k, and 200k).** Whereas, our approach is still better than all the variations, indicating that the performance gain indeed derives from the multi-task pre-training strategy. Besides, in our past experiments, we found that more pre-training steps are not always helpful to the downstream retrieval performance, as the model may overfit into the pre-training objective or even collapse. We show the MRR@10 performance on MS-MARCO dataset of two examples here: SimLM Retriever-1 w.r.t. pre-training steps: (40k: 37.6, 80k: 38.0, 120k: 38.0, 160k: 37.8, 200k: 37.7), a variation of our approach (Ours-w/o CPR): (40k: 37.9, 80k: 38.0, 120k: 37.9, 160k: 37.7, 200k: Collapsed). We can see that the performances of the baseline SimLM and Ours-w/o CPR get saturated after 80k pre-training steps, while more pre-training steps start to degrade the model performance and even make the model collapse.
>
> **C. We show the analysis about the learned "skills" in this part**. As the different objectives are used in the pre-training stage and unified into the bottleneck masked language model format, it is hard to explicitly show their endowed distinct ''skills'' on the dense retrieval models. The most straightforward ''skill'' is that they help our approach outperform competitive baselines and achieve new SOTA on multiple dense retrieval datasets in Table 2 and Table 3. The results in Table 5 have also shown that our approach can consistently outperform all baselines during the three-stage fine-tuning pipeline. Besides, as such a way can help our model pre-learn diverse knowledge from different tasks, it may be useful to improve our approach in other tasks (not just dense retrieval). To verify the hypothesis, we select the single-sentence and similarity tasks from the GLUE benchmark, which focus on predicting the acceptability, similarity, and paraphrase of sentences from different domains (e.g., news and misc). We conduct experiments by fine-tuning our model on these tasks, and show the performance of our approach and the original BERT in the following table, which has also been added in Appendix C.
> The CoLA task aims to judge whether a sentence is grammatical. MRPC, QQP, and STS-B are similarity and paraphrase tasks. As shown in Table, we can see that our approach can also improve the performance of BERT on these NLU tasks. It indicates that our multi-task pre-training indeed enriches the knowledge of the PLM and makes it generalize into other tasks.
>
> \\begin{array} {|l|c|c|c|c|}
> \\hline
>  & \\textrm{CoLA} & \\textrm{MRPC} & \\textrm{STS-B} & \\textrm{QQP} \\\\
> \\hline
> \\textrm{BERT} & 59.1 & 87.7 & 87.8 & 89.7  \\\\
> \\hline
> \\textrm{Ours}  & 60.7 & 89.1 & 88.0 & 89.8 \\\\
> \\hline
> \\end{array}

---

> > ### Comment · Reviewer_jK4d · 2022-11-25
> > **Response to author response**
> >
> > Thank you for the additional clarifications and results.
> >
> > I am quite happy with your response 1.B. and 2., which each clear up one of my major concerns. It seems that your claims are largely valid.
> >
> > While the claims seem well-supported, I am still not sure what a reader can learn from this paper. I am not quite convinced by responses 1.C. and 3. The paper combines a bunch of pretraining tasks, some of which are somewhat novel and each individually help, and shows some improvement over the SOTA. That is a nice engineering achievement, but how the individual pieces contribute to the overall model remains opaque. What are the central ideas in this paper and how can a reader two years from now apply these concepts to improve their existing IR system? In my opinion, this paper doesn't answer that question.
> >
> > I will nonetheless raise my score to reflect my increased confidence in the claims.

---

> > > ### Author Response · Authors · 2022-11-29
> > > **New Response to 3 and 1.C.**
> > >
> > > **We sincerely thank the reviewer for the updated comments and score. And we list our new response to 3 and 1.C with new experimental results and analysis.**
> > >
> > > **For 3, in this paper, the central ideas are as follows**:
> > > 1. We propose an effective framework that can unify a variety of available pre-training tasks for dense retrieval tasks into the bottlenecked masked autoencoding (BMAE) format.
> > > 2. We design MASTER, a multi-task pre-trained approach that integrates three types of pre-training tasks for pre-training, namely corrupted passages recovering, related passages recovering, and PLMs outputs recovering. The three types of tasks focus on modeling the semantic information within the passage, capturing the relationship among passages from the corpus, and learning the correlations between the passage and the PLM-generated related text, respectively.
> > > 3. Experimental results have shown the effectiveness of our approach, on four supervised text retrieval datasets (MS-MARCO, TREC-19/20, and NQ), a zero-shot retrieval benchmark consisting of 18 datasets (BEIR), and four natural language understanding tasks (CoLA, MRPC, STS-B, and QQP).
> > >
> > > According to the above central ideas, we show their possible usefulness for other information retrieval (IR) tasks.
> > > First, our proposed framework and the approach MASTER can provide valuable references for researchers to make use of the rich data correlations from the available data of other IR tasks, especially for the tasks that require modeling very informative data or capturing complex data correlations. E.g., for the personal recommendation task[1,2] where the given user context data has diverse formats (user's review texts and clicked item sequence), we can also unify them into the BMAE format for compressing the useful information from different data resources into the user representation. In this way, it is promising to pre-train more effective user representations that better depict the user characteristics for the recommendation task.
> > > Second, we have validated that the publicly released PLMs are also useful resources for IR tasks, and propose a novel way to use them, i.e., enriching the pre-training data. As these PLMs are pre-trained on the large-scale general corpus, they contain useful general knowledge beyond the downstream IR data. Therefore, using the available PLMs to enrich their pre-training data is capable of improving the generality of other IR models and alleviating the data sparsity problem.
> > > Third, we have released the code of our MASTER, and will also publicly release the checkpoints of our MASTER, to help researchers and developers quickly build effective IR systems.
> > >
> > > **For 1.C, to show how the individual pre-training task contributes to the model performance, we gradually add our proposed pre-training objectives (CPR, RPR, POR) to pre-train the model parameters**, namely PT with CPR, CPR+RPR, CPR+RPR+POR. After that, we fine-tune them and show the experimental results in the following table.
> > > \\begin{array} {|l|c|c|c|c|}
> > > \\hline
> > > & \\textrm{BERT} & \\textrm{PT w CPR} & \\textrm{PT w CPR+RPR} & \\textrm{PT w CPR+RPR+POR} \\\\
> > > \\hline
> > > \\textrm{Retriever}_1 & 32.9 & 37.2 & 37.6 & 38.3  \\\\
> > > \\hline
> > > \\textrm{Retriever}_2  & 34.0 & 39.6 & 39.8 & 40.4 \\\\
> > > \\hline
> > > \\end{array}
> > > From the Table, we can see that the performance is consistently increased by using more pre-training objectives. It indicates that more pre-training tasks are helpful for the dense retrieval task.
> > >
> > > [1] Chen, Li, Guanliang Chen, and Feng Wang. "Recommender systems based on user reviews: the state of the art." User Modeling and User-Adapted Interaction 25.2 (2015): 99-154.
> > >
> > > [2] Quadrana, Massimo, Paolo Cremonesi, and Dietmar Jannach. "Sequence-aware recommender systems." ACM Computing Surveys (CSUR) 51.4 (2018): 1-36.
> > >
> > > [3] Gao, Luyu, and Jamie Callan. "Condenser: a Pre-training Architecture for Dense Retrieval." Proceedings of the 2021 Conference on Empirical Methods in Natural Language Processing. 2021.

---

> > > ### Author Response · Authors · 2022-11-29
> > > **Additional Response about 1.C**
> > >
> > > **We sincerely thank the reviewer for the updated comments and score. And we list our response to 1.C with new case study.**
> > >
> > > To further analyze the effectiveness of them, we select representative examples from the MS-MARCO dataset, to show the Win/Lose case study of the retrieved top-1 document using different pre-training objectives.
> > > \\begin{array} {|l|l|}
> > > \\hline
> > > & \\textbf{Query}: \\textrm{perveance definition} \\\\
> > > \\hline
> > > \\textrm{BERT} & \\textrm{\\textbf{Document}: Definition of Pervade. transitive verb To pass or flow through, as an aperture, pore, or interstice; to permeate.} \\\\
> > > \\hline
> > > \\textrm{PT w CPR} & \\textrm{\\textbf{Document}: Perveance. Perveance is a notion used in the description of charged particle beams. The value of perveance indicates how significant the space charge effect is on the beam motion.} \\\\
> > > \\hline
> > > \\end{array}
> > > From the above Table, we can see that given the query about the definition of perveance, the original BERT has retrieved the definition of pervade in the 1st place by mistake.  As discussed in Condenser[1], the internal attention structure of BERT is not ready-to-use for dense encoders, which fails to aggregate the information of the keyword "perveance" into the dense representation. As a comparison, after pre-training with CPR that focuses on modeling the semantic information within the passage, our model successfully retrieves the relevant document.
> > > \\begin{array} {|l|l|}
> > > \\hline
> > > & \\textbf{Query}: \\textrm{when typing which formula is used to measure accuracy} \\\\
> > > \\hline
> > > \\textrm{PT w CPR} & \\textrm{\\textbf{Document}: Formula for Measure Typing Speed in Mastering Typing. Include gross WPM and net WPM that used to calculate accuracy in Mastering Typing Formula for Measure Typing Speed in Mastering Typing.} \\\\
> > > \\hline
> > > \\textrm{PT w CPR+RPR} & \\textrm{\\textbf{Document}: Formula to measure typing speed in WPM. Accuracy Formula: Accuracy is a percentage ratio of Gross and Net Word Speed: Accuracy = Net-WPM / Gross-WPM * 100. Calculation of Errors. Errors are calculated by following two criteria. Errors that are made and corrected; Errors that are made and not corrected.} \\\\
> > > \\hline
> > > \\end{array}
> > > From the above Table, we can see that the retrieved documents from the two variations are indeed relevant to the given query, and many words are co-occurring in the query and documents. Whereas, the retrieved one from PT w CPR+RPR is more proper to answer the given query, since it focuses on the keyword "measure accuracy" and provides a detailed description of the "Accuracy Formula" to illustrate how it can measure the accuracy. It indicates that with the help of the RPR task, our model can better focus on useful information from the text that can capture important relationships between queries and passages.
> > > \\begin{array} {|l|l|}
> > > \\hline
> > > & \\textbf{Query}: \\textrm{what does dress it down mean} \\\\
> > > \\hline
> > > \\textrm{PT w CPR+RPR} & \\textrm{\\textbf{Document}: When people say dress down/up what does it mean?. Asker's rating. 1  Dress Down Meaning. 2  I can nest answer this with an example. 3  to dress it up means to go more formal than school/work clothes. 4  Dressing up is when you are doing something fancier then an everyday thing. 5  For the best answers, search on this site.} \\\\
> > > \\hline
> > > \\textrm{PT w CPR+RPR+POR} & \\textrm{\\textbf{Document}: When people say dress down/up what does it mean?. Best Answer: Dress up means to dress more fancier, like you are going out to a fancy restaurant and you want to dress up for it. To dress down, means if you are wearing something fancy and you are going somewhere that doesn't need you to be so well dressed, you dress down into something more casual.} \\\\
> > > \\hline
> > > \\end{array}
> > > From the Table, we can see that the two retrieved documents are very relevant to the given query and both focus on the keyword "dress down". However, the retrieved one from PT w CPR+RPR does not clearly answer the given query but just lists several possible answers. As a comparison, the retrieved one from PT w CPR+RPR+POR provides a concrete answer about the means of "dress down". Since POR loss aims to learn the correlations between the passage and the PLM-generated related text, it can meet more examples from other publicly released PLMs, further improving its capacity to model complex query-document relationships.

---

> > > ### Author Response · Authors · 2022-12-07
> > > **A Kind Remind**
> > >
> > > Dear Reviewer jK4d,
> > >
> > > Thanks for your kindly updated scores and insightful suggestions of our paper. We have tried our best to elaborate your mentioned unclear points, i.e., no qualitative analysis that shows us that the different pretraining objectives actually teach the model distinct "skills".
> > >
> > > Concretely, we conduct new experiments as **the quantitative analysis to show the effectiveness of different pre-training objectives**, and also perform **the qualitative analysis by showing their Win/Lose case study** of the retrieved top-1 document. Besides, we have **released our code, pre-training data and checkpoints in the anonymous Webpage**, i.e.,https://anonymous.4open.science/r/MASTER-FF38/README.md, and provided detailed command lines for using them to implement our pre-training process or fine-tuning.
> > >
> > > We would like to know whether you find our response satisfactory, or if there are more questions that we could clarify. Since the rebuttal stage is coming to an end, we are more than happy to hear your comments and address any of your further concerns during the remaining time.
> > > Best,
> > > Authors

---

> ### Author Response · Authors · 2022-11-19
> **Response to the Concerns about the Comparison to SimLM, Limited Novelty and Reproducibility**
>
> Thanks for your insightful suggestions and we have listed our response to the concerns about the comparison to SimLM, limited novelty and reproducibility. If you still have any other questions, please do not hesitate to tell us. We will continue to try our best to answer for you.
>
> **2. For the question about comparisons to SimLM**: As shown in General Response, we have considered this issue before and guarantee that most of the pre-training and fine-tuning settings are the same as SimLM for a fair comparison. Experimental results in Table 6 have shown that our approach can consistently outperform SimLM under different fine-tuning settings.
>
> **3. For the question about limited novelty**: for the dense retrieval task, a surge of pre-training strategies, e.g., weak-supervised contrastive learning and masked autoencoding, have been proposed. Although each dividual pre-training strategy can significantly boost the model performance, due to the divergence of different training objectives and data formats, it is hard to integrate all these tasks to pre-train a more effective PLM for the dense retrieval task. Empirically, we have conducted experiments by simply combining several commonly-used pre-training tasks (contrastive learning like Cocondenser and bottleneck MLM like SimLM), and see that the model does not perform well (Retriever-1: 36.6), compared with SimLM that only adopts the bottleneck MLM (Retriever-1: 38.0).  In this work, our major contribution is to propose an effective framework that can successfully unify and integrate the above available tasks to improve the pre-training for dense retrieval. Instead of simply combining the bottlenecked model architecture and multi-task pre-training, we carefully extract and categorize the useful semantics and relations hidden in all available pre-training tasks (e.g., co-occurrence relation and keywords) and finally determine three types of pre-training tasks for learning, covering most of the useful semantics and relations that have shown effective in existing works. Based on them, we also adjust the model architecture incorporating one deep encoder and multiple shallow decoders to construct multiple information bottlenecks that force the encoder to generate high-quality and informative dense representations. Furthermore, we also consider capturing useful semantics and relations beyond the pre-training corpus, which have not been well considered in existing works. We design the PLMs outputs recovering task, which forces the dense vector of the input passage to recover the output text from external public generative PLMs (i.e., GPT-2 and DocT5). Such a way is similar to the knowledge distillation process that transfers the learned knowledge from these external generative PLMs into our model, further enriching the information of the dense vector.
>
> **4. For the reproducibility question**: We have prepared our approach's code and uploaded it to the supplementary material. And we will release our pre-trained checkpoints after the review period to guarantee that everyone can re-implement the experimental results in our paper.

---

### Official Review · Reviewer_JgYb · 2022-10-26

**Confidence:** 3
**Correctness:** 4
**Technical Novelty And Significance:** 3
**Empirical Novelty And Significance:** 3
**Recommendation:** 6

**Clarity, Quality, Novelty And Reproducibility:**

The paper is clear and easy to follow.

Minor suggestions:
1. Section 4, "Approach". "an approach to pre-training an effective ..." -> "pre-train"?


**Strength And Weaknesses:**

Strengths
1. The paper proposed integrating multiple pre-training tasks via a unified text-to-text format. The bottlenecked masked autoencoder also allows task-specific decoders.
2. The results are competitive.


**Summary Of The Paper:**

The paper proposed multiple pre-training tasks to be used with the bottlenecked masked autoencoder architecture, designed specifically for dense retrieval. The three types of tasks are corrupted passages recovering, related passages recovering, and pre-trained language model (PLM) outputs recovering. The tasks are integrated by formulating each tasks in a unified text-to-text format. Each pre-training task has its own task-specific shallow decoder. The claimed benefits is that the Transformer encoder is forced to compress the information into a dense vector. The empirical results show the proposed method outperforms competitive baselines.

**Summary Of The Review:**

The paper proposed carefully designed pre-training tasks, a way to integrate multiple pre-training tasks and show competitive results empirically.

---

> ### Author Response · Authors · 2022-11-19
> **Response to the Minor Suggestions**
>
> Thanks for your suggestions and appreciate your positive score. We have modified the mentioned typo, and have carefully read and checked the writing of our manuscript. If you have any other questions, please feel free to ask us. We will try our best to answer for you.

---

### Author Response · Authors · 2022-11-19
**The Detailed Experimental Settings of our Approach and Ablation Study**

Thanks for the constructive suggestions and the valuable time of the reviewers.
Here, I will introduce the experimental setting of our approach, as it may not be clear in our last manuscript due to the page limitation.
1. **Implementation details of our approach**: Most of the hyper-parameters in the pre-training stage of our approach are the same as SimLM. Concretely, the batch size is 2048, the learning rate is 3e-4, and the mask rate of the encoder and decoders are 30% and 50%, respectively. The pre-training step is set to 120k. During fine-tuning, we also follow the same hyper-parameters setting as SimLM (more details can be found in Table 13 of SimLM). In this way, we can guarantee a fair comparison between our approach and the previous SOTA method SimLM. The experimental results in Table 6 show that our approach can outperform SimLM during the three-stage fine-tuning pipeline using different negatives and labels, i.e., Retriever-1, Retriever-2, and Retriever-distil. Besides, our method can get better results if we simply adjust the hyper-parameters during fine-tuning, i.e., learning rate 5e-6 and negative rate 1:23. We list the new results on MS-MARCO-Passage dataset as follows, and have updated them in our new manuscript.

\\begin{array} {|l|c|c|}
\\hline
& \\textrm{SimLM} & \\textrm{Ours} \\\\
& \\textrm{MRR}@10, \\textrm{R}@1k & \\textrm{MRR}@10, \\textrm{R}@1k \\\\
\\hline
\\textrm{Retriever}_1 & 38.0,  98.3 & 38.3, 98.8  \\\\
\\hline
\\textrm{Retriever}_2  & 39.1,  98.6 & 40.4, 98.8 \\\\
\\hline
\\textrm{Retriever}_\\textrm{distil}  & 41.1,  98.7 & 41.5, 98.8 \\\\
\\hline
\\end{array}

2. **Implementation details of ablation study**: For all the variations of our approach, we have pre-trained them for 200k steps since the pre-trained losses generally do not decrease, or the model may collapse after 200k steps. Then, we select the pre-trained checkpoints in the steps of 80k, 120k, 160k, and 200k for fine-tuning and report the best performance in Table 5. Following the reviewers' suggestion, we further fine-tune the variations (including our approach) with their best checkpoints by three times using different random seeds to better show the effectiveness of our multi-task pre-training. We report their means and variances on MS-MARCO-Passage dataset as follows.


\\begin{array} {|l|c|c|c|c|c|}
\\hline
\\textrm{MRR}@10 & \\textrm{Ours} & \\textrm{w/o CPR} & \\textrm{w/o RPR} & \\textrm{w/o POR} & \\textrm{+Shared-Dec} \\\\
\\hline
\\textrm{Retriever}_1 & 38.3 (0.016) &  37.7 (0.016) & 37.6 (0.007) & 37.6 (0.016) & 37.4 (0.016)  \\\\
\\hline
\\textrm{Retriever}_2  & 40.4 (0.016) &  39.9 (0.007) & 39.8 (0.009) & 39.8 (0.002) & 39.1 (0.069) \\\\
\\hline
\\end{array}

---

### Author Response · Authors · 2022-11-19
**New Revision of our Manuscript**

We sincerely thank the three reviewers for their insightful and constructive feedback. We have provided a separate response to each reviewer, and also updated the paper following the revision suggestions of the reviewers. We list the main revision content as follows:

1. **We add the zero-shot experiments on BEIR benchmark in Section 5.2, and show the experimental results on Table 4.** We can see that our approach can mostly outperform competitive baselines in the 18 datasets.

2. **We add the evaluation on four natural language understanding tasks in Appendix C, and show the results on Table 8.** We can see that our pre-training approach can also benefit for these NLU tasks, showing the learned distinct skills of our multi-task pre-training.

3. **We re-conduct the ablation study using different random seeds in Table 6.** We fine-tune the variations (including our approach) with their best checkpoints by three times using different random seeds, and report their means and variances on MS-MARCO-Passage dataset. We can see that removing each pre-training objective would lead to performance degradation, and their low variances (<0.07) indicate that each objective stably improves the model performance.

4. **We fix the mentioned typos, carefully read and revise the writing of our paper.**

5. **For reproducibility, we upload the code of our approach in the supplementary material.**

---

### Author Response · Authors · 2022-12-11
**Reminder for the Discussion**

Dear Reviewers,

We want to send you a friendly reminder that the second stage of discussion will be complete soon. Here are the things that we have added and resolved by your valuable feedback!

1. **New zero-shot experiments on BEIR benchmark**.
2. **New evaluation on four natural language understanding tasks**.
3. **The case study of our approach compared with baselines**.
4. **The re-conducted ablation study using different random seeds**.
5. **The released code, data, and model parameters of our approach in https://anonymous.4open.science/r/MASTER-FF38/README.md.**
6. **We fix the mentioned typos, carefully read and revise the writing of our paper.**

Thanks for your willingness to reconsider your score based on our responses, and we really want to know whether our responses address your concerns. If there is any other concern that we could not address in the response, please feel free to let us know, and we would be happy to provide further explanation.

Thanks

---

### Decision · Program_Chairs · 2023-01-20

**Decision:**

Reject

**Justification For Why Not Higher Score:**

I think that the limited technical novelty compared to the baseline, and the limited improvements obtained by the method (esp. on the BEIR benchmark), do not justify the acceptance of this paper.

**Justification For Why Not Lower Score:**

N/A

**Metareview: Summary, Strengths And Weaknesses:**

The authors propose a new pre-training method for dense retrievers, based on a bottlenecked masked auto-encoder. This allows to unify multiple pre-training tasks, using a text-to-text framework, where a different decoder is used for each task. The proposed method is evaluated on multiple benchmarks, such as MS-MARCO, TREC and BEIR.

Overall, the reviews for this paper are borderline. The reviewers like the fact that the proposed method is a simple way to unify different pre-training schemes for retrieval, using a text-to-text framework. On the other hand, the reviewers also believed that the method is quite similar to the SimLM baseline. Moreover, the improvements over baselines are quite small, and some reviewers believe that the ablations are not sufficient to motivate what technical contributions are responsible for the improvements. Given that other works also improve the SimLM baseline, the reviewers feel that this paper needs major revision before acceptance. I tend to share this conclusion, and thus recommend to reject the paper.